# Rolling Bearing Remaining Useful Life Prediction Based on CNN-VAE-MBiLSTM

**DOI:** 10.3390/s24102992

**Published:** 2024-05-08

**Authors:** Lei Yang, Yibo Jiang, Kang Zeng, Tao Peng

**Affiliations:** 1The ZJU-Hangzhou Global Scientific and Technological Innovation Center, Zhejiang University, Hangzhou 311215, China; 2The Jiaxing Shutuo Technology Co., Ltd., Jiaxing 314031, China; jyb892902296@sina.com; 3The Zhejiang Loong Airlines Co., Ltd., Hangzhou 311243, China; kang.zeng@loongair.cn; 4State Key Laboratory of Fluid Power Components and Mechatronic Systems, School of Mechanical Engineering, Zhejiang University, Hangzhou 310058, China; tao_peng@zju.edu.cn

**Keywords:** remaining useful life, variational autoencoder, bi-directional long short-term memory

## Abstract

Ensuring precise prediction of the remaining useful life (RUL) for bearings in rolling machinery is crucial for preventing sudden machine failures and optimizing equipment maintenance strategies. Since the significant interference encountered in real industrial environments and the high complexity of the machining process, accurate and robust RUL prediction of rolling bearings is of tremendous research importance. Hence, a novel RUL prediction model called CNN-VAE-MBiLSTM is proposed in this paper by integrating advantages of convolutional neural network (CNN), variational autoencoder (VAE), and multiple bi-directional long short-term memory (MBiLSTM). The proposed approach includes a CNN-VAE model and a MBiLSTM model. The CNN-VAE model performs well for automatically extracting low-dimensional features from time–frequency spectrum of multi-axis signals, which simplifies the construction of features and minimizes the subjective bias of designers. Based on these features, the MBiLSTM model achieves a commendable performance in the prediction of RUL for bearings, which independently captures sequential characteristics of features in each axis and further obtains differences among multi-axis features. The performance of the proposed approach is validated through an industrial case, and the result indicates that it exhibits a higher accuracy and a better anti-noise capacity in RUL predictions than comparable methods.

## 1. Introduction

Bearings are a fundamental and vulnerable part of industrial rotating equipment. Numerous factors (e.g., running load, operation temperature, lubrication, installation, corrosion, material defects, etc.) lead to severe bearing faults and influence the normal operation of machines [1]. Thus, regular bearing maintenance is crucial for reducing machine downtime and improving productivity [2]. In recent years, predictive maintenance (PdM) has become an increasingly significant field in modern manufacturing, as it can estimate the health status of machines to minimize risks of sudden breakdowns [3,4]. One key technique of PdM is RUL prediction that refers to the remaining time left for machines to operate normally before a serious bearing failure occurs [5]. Consequently, accurate RUL prediction of bearings contributes to minimizing machine downtime, reducing maintenance frequency, and maximizing the service life of bearings [6,7,8,9].

Generally, approaches for predicting the RUL of bearings can be roughly classified as two main groups: mechanism model methods [10,11,12] and data-driven methods [13,14,15]. Specifically, mechanism model methods typically rely on failure principles of bearings and try to establish accurate mathematical models to describe the degradation process of bearings [16]. Hu et al. [17] proposed an RUL prediction model for bearings based on the diffusion process; the model addresses the uncertainty in prediction results and enhances the accuracy of predictions. Gao et al. [18] developed fatigue reliability models that consider the combined effects of fatigue damage accumulation and effective stress growth, resulting in an accurate performance degradation prediction of composite materials. In theory, mechanism model methods have the potential to adequately reflect the system nature by describing the mechanisms and characteristics of the bearing degradation process [19]. Nevertheless, for complex mechanical systems, physical principles underlying bearing failures are not yet fully understood [20], which leads to difficulties in developing precise and reliable mechanism models.

Recently, the development of artificial intelligence (AI) as well as big data technologies brings significant opportunities for data-driven methods to eliminate the need for complex control equations during the analysis process [21]. By extracting features from enormous data and establishing specific relational models between feature patterns and the RUL of bearings, data-driven methods offer innovative solutions in the prediction of RUL for bearings [22]. In the literature [23], based on the open-accessed bearing dataset from the FEMTO-ST institute, principal component analysis (PCA) as well as least squares-support vector regression (LS-SVR) are applied to extract features and predicting RUL, respectively. This method provides a soft computing technique to identify patterns among features, which improves the accuracy in RUL prediction. Additionally, Dong et al. [24] proposed a novel method that integrates kernel PCA with a support vector machine (SVM). In this method, vibration signals are decomposed to obtain fault information. Following that, characteristic features extracted by a kernel PCA are inputted into the SVM to establish a classification model for operational status. Ultimately, this approach achieves effective recognition of bearing operating status.

Currently, deep learning (DL) is considered as a major breakthrough in the data-driven methods. With deep neural networks, DL is able to capture deep representation of the dataset and achieve better performance than other data-driven methods in the fault diagnosis and the prediction of RUL [25,26,27]. Gao et al. [28] integrate fuzzy inference and neural networks to capture the nonlinear relationship between parameters and fatigue status. Additionally, introducing non-proportionality and phase differences enables the model to accurately predict the fatigue life of various materials. Zhu et al. [29] combined time–frequency representations (TFRs) and multiscale convolutional neural network (MSCNN) for bearing RUL prediction, in which wavelet transform (WT) is utilized to obtain non-stationary property of TFRs and address the difficulty in applying CNN directly to raw time series. Xiao et al. [30] proposed a fusion method that merges empirical mode decomposition with a gated recurrent unit (GRU) to effectively address the problem of accurately assessing bearing degradation. The key innovation of this method lies in decomposing the original signal and extracting the most sensitive trend features, which are then inputted into GRU to calculate the health index.

In practical industrial environments, the large data volume, high data dimension, strong interference noise, and coupling effects between parameters make it difficult to achieve high accuracy and strong robustness in the prediction of bearing RUL. Hence, a novel prediction framework called CNN-VAE-MBiLSTM is proposed for RUL prediction of bearings in this paper. The CNN-VAE part of the framework is obtained by fusing symmetric CNN and VAE; it can effectively capture accurate low-dimensional TFRs from time–frequency spectrum of signals using the advantage of efficient image processing provided by CNN and the continuous learning ability of data distribution offered by VAE. Then, the MBiLSTM is introduced to transform features from multi-axis signals into estimated RUL values. The MBiLSTM consists of two steps: Firstly, statistical variables and TFRs from each direction are input into a sub-model to encode temporal information. In the second step, outputs of sub-models are combined to extract the differences among multi-axis features.

At last, the effectiveness of CNN-VAE-MBiLSTM is verified using experimental datasets of bearings. The key contributions of this study are presented as follows:(1)The CNN-VAE part is an unsupervised model that can adaptively extract TFRs without relying on hand-designed labels, which avoids laborious work of feature construction, eliminates the influence of personal participation, and successfully applies the high-dimensional time–frequency spectrum to RUL prediction.(2)Bi-directional long short-term memory (BiLSTM) is employed as the sub-model in MBiLSTM, which is excellent for capturing sequential characteristics of features and has a significant improvement in accuracy of RUL prediction. In addition, the two-step approach designed in MBiLSTM imitates the architecture of ensemble learning to enhance the accuracy and robustness of RUL prediction. Experimental results indicate that the MBiLSTM has better performance than the single BiLSTM.

Subsequent sections of this paper are arranged as follows. The related works are reviewed in Section 2. Subsequently, problem formulation and methodology are described in Section 3. Then, the proposed approach is verified in Section 4, which consists of dataset description, evaluation metrics, feature construction, as well as the RUL prediction and discussion. Lastly, the conclusion of the paper is offered in Section 5.

## 2. Related Work

Severe bearing failures will result in equipment breakdowns, leading to substantial economic loss and threatening the health of operators [31]. Therefore, timely analysis of operation conditions for bearings is of great research importance. Recent studies have demonstrated the effectiveness of using operating data to reflect the bearing degradation caused by material defects or other intricate factors [32]. Therefore, data-driven approaches have become an essential strategy of the PdM for bearings.

Typically, the construction of RUL prediction models for bearings using data-driven approaches consists of two key phases: one is to utilize signal processing techniques (SPTs) to extract physical features, and the other is to employ machine learning models to learn the underlying correlation between these features and bearing degradation [33,34]. For instance, Singleton et al. [35] extracted the time–frequency domain features (TFDFs) from vibration signals and then tracked the TFDF to evaluate the RUL of bearings using curve fitting and extended Kalman filtering algorithms. Huang et al. [36] integrated the attention mechanism into neural network and utilized time domain features (TDFs) and frequency domain features (FDFs) as inputs, achieving good RUL prediction results of bearings.

Although physical features extracted using SPTs have proven effective in qualitative classification of bearing health status, these features still face challenges in fully capturing subtle changes during the degradation process for quantitative prediction [37]. Moreover, due to the diverse and complex nature of degradation processes, determining appropriate features also requires substantial expertise and human labor, so that some researchers introduce end-to-end deep frameworks into RUL estimation in bearings [38,39]. In the literature [40], the LSTM network is combined with the CNN network to form an end-to-end deep framework. Within this framework, the convolutional layer directly extracts degradation features from sensor data, while LSTM layers are utilized for accurate quantitative prediction of the degradation process. In addition, Ye et al. [41] adopted multi-scale convolutional autoencoder (MSCAE) to automatically capture both global and local information from vibration signals. Health indicators (HI) were then constructed to replace time–frequency features as inputs of the prediction model (i.e., LSTM network). Ultimately, the effectiveness of this approach was validated using an open-source dataset. Compared to machine learning models, end-to-end deep framework not only significantly improves prediction performance but also simplifies the modeling process by skipping the feature engineering. However, the simplification of processes results in redundancy of hyper-parameters and poor model generalization [42].

To solve the aforementioned issues, there is an increasing interest to introduce deep learning methods into feature engineering and model construction, respectively. Specifically, traditional SPTs are combined with deep neural networks to extract deep features. Then, these deep features are used in a deep neural network to establish an accurate relationship with the target values [43]. Li et al. [44] proposed an intelligent method for RUL prediction based on deep CNNs, where short-time Fourier transform (STFT) is employed to obtain the time–frequency spectrum of vibration signals. The time–frequency spectrum is then processed by CNN to extract and analyze multi-scale features, resulting in high-precision RUL prediction results. Saucedo-Dorantes et al. [45] introduced a novel data-driven diagnosis methodology for identifying bearing faults, in which stacked auto-encoders (SAE) are used to extract fault-related deep features and a deep neural network is employed to fuse the information from different domains. The experiment showed that this approach achieves advantageous results in the fault diagnosis of different bearings.

As mentioned above, deep neural networks have been explored to predict bearing RUL in some research, but further improvements are needed to achieve more accurate and robust predictive performance.

## 3. Problem and Methods

Precise RUL prediction plays a pivotal role in the PdM for bearings, serving as a vital measure to improve bearing utilization efficiency and prevent failures of machines. The diagram of the proposed approach is displayed in Figure 1, and the RUL prediction of bearings consists of three parts.

(1)Data Acquisition and Pre-processing: Generally, the sensor data for bearings includes a vibration signal [46], acoustic emission signal [47], and temperature signal [48]. Among them, vibration-based techniques have been widely acknowledged as some of the most effective approaches to monitor the degradation of bearings [49]. Consequently, this paper applied vibration signal to predicting the RUL of bearings. Following that, to control the data size of signal, a sliding window technique was used to divide the raw signal into several segments. It is worth noting that the sliding window size and sliding amount should be determined by the specific industrial process and the characteristic of the machine.(2)Feature Engineering: In this phase, representative features related to bearing health status were extracted from sensor data. The representative features included TDFs, FDFs, and TFDFs, in which TDFs and FDFs are obtained with traditional SPTs, while TFDFs are attained with the CNN-VAE part of the proposed approach.(3)Model Construction: The MBiLSTM part of the proposed approach was constructed to establish the underlying correlation between these representative features and the RUL of bearings. Furthermore, capturing sequential characteristics and extracting differences among multi-axis information make the MBiLSTM achieve high performance in RUL prediction.

As noted above, the details of feature engineering and model construction are discussed in subsequent subsections.

### 3.1. Feature Engineering

SPTs have the advantage of extracting physically interpretable features from high-frequency vibration signals. However, these techniques still face challenges when it comes to fully interpreting and compressing the valuable information of raw signals [25]. To maximize the mining of effective information and enhance the robustness of prediction model, SPTs were combined with deep learning methods in the proposed approach, achieving the extraction of following features:

#### 3.1.1. TDFs

TDFs of vibration signals focus on estimating the variation of amplitudes over time, which are effective at capturing the decay trend of bearings. Classically, TDFs comprise 16 statistical parameters, as illustrated in Table 1, where $x_{i}$ is the *i*th amplitude value in vibration signal, and *N* denotes the data length of vibration signal.

#### 3.1.2. FDFs

Using discrete Fourier transform technology [50], vibration signals can be decomposed into a linear combination of multi-sinusoidal waves, as defined by:(1)$$X\left(t\right)=\sum_{w=0}^{\infty }a_{n}cos\left(2\pi w*t\right)+b_{n}sin\left(2\pi w*t\right)$$
where $X\left(t\right)$ is raw vibration signal, $a_{n}$ and $b_{n}$ are weights of multi-sinusoidal waves, and w is the frequency of sinusoidal wave.

During the above process, types and weights of these waves are sensitive to the differences in vibration signals over time. Hence, FDFs are essential in the analysis of RUL for bearings. Specifically, FDFs consist of 12 statistical parameters, as described in Table 2. In Table 2, $w_{i}$ and $f_{i}$ represent the weight and frequency of *i*th sinusoidal wave, and M is the number of sinusoidal waves.

Given that the extraction of TDFs and FDFs from each vibration signal results in an exponential growth in the feature number, feature evaluation is further utilized to select effective features and reduce the feature number. The feature evaluation includes monotonicity evaluation, correlation evaluation, and robustness evaluation. The formulas are expressed as follows:(2)$$S_{Mon}=\frac{\left|\sum_{n=1}^{N}H\left(x_{n+1}-x_{n}\right)-\sum_{n=1}^{N}H\left(x_{n}-x_{n-1}\right)\right|}{N-1}$$
(3)$$H(x)=\left\{\begin{array}{c} 0,\;\;x<0\\ 1,\;\;x\geq 0 \end{array}\right.$$
(4)$$S_{cor}=\frac{\left|\sum_{n=1}^{N}\left(x_{n}-\bar{x}\right)*\left({rul}_{n}-\bar{rul}\right)\right|}{\sqrt{\sum_{n=1}^{N}{\left(x_{n}-\bar{x}\right)}^{2}*\sum_{n=1}^{N}{\left({rul}_{n}-\bar{rul}\right)}^{2}}}$$
(5)$$S_{rob}=\frac{1}{N}\sum_{n=1}^{N}exp\left(-\left|\frac{x_{n}-b_{n}}{x_{n}}\right|\right)$$
where N denotes the sequence length of feature, $x_{n}$ denotes the *n*th value of feature, $b_{n}$ denotes the *n*th stable value of feature, ${rul}_{n}$ denotes the *n*th value of RUL, and $S_{Mon}$, $S_{cor}$, and $S_{rob}$ represent scores of monotonicity evaluation, correlation evaluation, and robustness evaluation, respectively.

#### 3.1.3. TFDFs

Vibration signals of bearings in industrial equipment are non-stationary signals, while discrete Fourier transform technology treats the signal as a stationary signal when extracting FDF, and results in the loss of temporal information in the original signal. To address this, various time–frequency analysis methods have been developed to extract frequency spectra with temporal distribution information, including STFT [51], WT [52], Hilbert–Huang transform (HHT) [53], etc. Among them, the WT and the HHT involve the decomposition of signal and the filtration of sub-signals, requiring hand-designed parameters or artificial selections. Hence, this study combines STFT with a deep neural network to achieve automatic distribution learning and compression for TFDFs.

The schematic diagram of TFDFs extraction is shown in Figure 2 and the calculation steps are described as follows.

Firstly, the raw vibration signals are transformed into 2D images of time–frequency spectrum using the STFT. The definition of the STFT is expressed as Equation (6):(6)$$S\left(\tau ,\omega \right)=\frac{1}{2\pi }\int_{-\infty }^{+\infty }v\left(t\right)\eta \left(t-\tau \right)e^{-j\omega t}dt$$
where $v\left(t\right)$ is raw vibration signal, $\eta \left(t\right)$ is a symmetric window function [54], and $S\left(\tau ,\omega \right)$ denotes the function of both modulated frequency (ω) and translated time (τ).

Secondly, 2D images of the time–frequency spectrum provide a comprehensive representation of the raw vibration signal, while extremely high dimensionality make them hard to directly serve as the input of RUL prediction model for bearings. Therefore, further information compression is needed. As a method of unsupervised learning, VAE has found extensive application in feature dimensionality reduction tasks, which can effectively estimate numerous high-dimensional samples and produce low-dimensional representations with continuity [55,56,57]. Additionally, CNN has been widely employed in the field of image processing, exhibiting significant achievements in extracting texture, contour, and semantic features from images [58,59,60]. Consequently, a novel feature extraction model named CNN-VAE is proposed, combining strengths of the VAE and the CNN. The framework of CNN-VAE comprises two symmetric subnetworks: the encoding block and the decoding block. Each block consists of several convolutional layers and fully connected layers. The training process of the CNN-VAE involves five steps:(1)Feature Encoding: The original 2D time–frequency spectrum is processed through convolutional layers at first, resulting in a down-sampled image achieved by multiple convolutions and pooling operations. The down-sampled image is then flattened into a 1D vector and passed through multiple fully connected layers to further reduce dimensionality. This process ultimately yields the TFDF and its standard deviations.(2)Resampling: Treating TFDFs and their standard deviations as the mean and standard deviation of a normal distribution, new low-dimensional samples are generated by randomly sampling from this distribution. These samples serve as inputs to the decoding block.(3)Feature Decoding: Utilizing multiple fully connected layers and transposed convolutional layers [61], the low-dimensional samples are up-sampled to reconstruct a 2D time–frequency spectrum.(4)Hyperparameter Update: The difference between the reconstructed 2D time–frequency spectrum and the original 2D time–frequency spectrum is used to calculate the reconstruction error. This error is then used to update the hyperparameters of the CNN-VAE with the error backpropagation algorithm [62].(5)Steps 1 to 4 are repeated for several epochs until the reconstruction error gradually stabilizes to a sufficiently low value.

Finally, the encoding block of the CNN-VAE was utilized to obtain the TFDFs.

### 3.2. Model Construction

In many industrial cases, bearings gradually degrade rather than abruptly failing in their life cycles. Therefore, in addition to independently assessing the operational features of bearings collected at each moment, it is equally crucial to examine feature differences between adjacent moments.

The recurrent neural network (RNN) is specifically designed for handling sequential data, as illustrated in Figure 3. Features from different time steps are sequentially fed into the network in accordance with the time order to update the current state of network, enabling the RNN to capture temporal dependencies within the sequence [63,64]. However, the problems of vanishing and exploding gradients impose limitations on the performance of RNN when handling long time-series data [65,66]. To overcome these challenges, a novel type of RNN called the BiLSTM has been developed, which consists of two parallel layers that operate in both forward and backward propagation directions [67]. The structure of the BiLSTM is depicted in Figure 4, and corresponding operations are described in Equations (7)–(14).
(7)$$\sigma \left(x\right)=1/\left(1+e^{1-x}\right)$$
(8)$$tanh\left(x\right)=\left(e^{x}-e^{-x}\right)/\left(e^{x}+e^{-x}\right)$$
(9)$$f=\sigma \left({input}_{t}W_{input}^{f}+h_{t-1}W_{h}^{f}+b^{f}\right)$$
(10)$$g=tanh\left({input}_{t}W_{input}^{g}+h_{t-1}W_{h}^{g}+b^{g}\right)$$
(11)$$i=\sigma \left({input}_{t}W_{input}^{i}+h_{t-1}W_{h}^{i}+b^{i}\right)$$
(12)$$o=\sigma \left({input}_{t}W_{input}^{o}+h_{t-1}W_{h}^{o}+b^{o}\right)$$
(13)$$c_{t}=f⊙c_{t-1}+g⊙i$$
(14)$$h_{t}=o⊙tanh\left(c_{t}\right)$$
where ${input}_{t}$, $h_{t}$, and $c_{t}$ denote the input vector, hidden state of neuron and memory of neuron at time step t, respectively; $W_{input}^{f}$, $W_{h}^{f}$, $W_{input}^{g}$, $W_{h}^{g}$, $W_{input}^{i}$, $W_{h}^{i}$, $W_{input}^{o}$, and $W_{h}^{o}$ are weight matrices and $b^{f}$, $b^{g}$, $b^{i}$, and $b^{o}$ are vectors of bias; and ⊙ means the point-wise multiplication of two vectors.

Based on the utilization of feature engineering and the BiLSTM network, a MBiLSTM is presented to predict the RUL of bearings. For better understanding, the whole proposed deep learning framework will be explained below. The schematic diagram of CNN-VAE-MBiLSTM is shown in Figure 5, and the calculation steps are outlined as follows:(1)Automatic Extraction of Sequential Characteristics: Using the sliding window technique, original vibration signals are divided into numerous signal segments. Then, TDFs, FDFs, and TFDFs extracted from each signal segment are normalized and employed as inputs for BiLSTM, aiming at deep encoding of degradation information and automatic extraction of sequential characteristics.(2)Fusion of Multi-Axis Information: Typically, vibration signal acquisition involves multiple directions, so the outputs of the BiLSTM for each vibration direction will be concatenated into a matrix. This matrix is subsequently fed into another BiLSTM to fuse information, achieving the adaptive exploration of trend differences across multiple vibration directions.(3)RUL Estimation: To enhance the robustness of prediction, the sliding window in (1) adopts an overlapping sliding mechanism, allowing the RUL value at each moment to be estimated multiple times. Ultimately, the median of the multiple estimated values is taken as the final result for reducing the impact of random errors in prediction results.

## 4. Experiment and Analysis

The proposed RUL prediction approach for bearings was validated using an industrial case from a textile company located in Jinhua, Zhejiang Province, China. In this case, based on the prior maintenance knowledge of machines, bearing datasets were simultaneously collected from the crucial bearing installed in six weaving machines (Picanol GTMax-I 3.0,Picanol Group, Ieper, Belgium), and these weaving machines kept intermittently operating until they had to be shut down due to severe bearing failures. In addition, only vibration signals were recorded in bearing datasets using sensors (CT1010L and PCIE-1803, Shenzhen Jilantin Intelligent Technology Co., Ltd., Shenzhen, China). The experimental platform is shown in Figure 6.

### 4.1. Dataset Description

During the entire lifespan of the weaving machine, vibration signals of main bearing contained multiple axes, namely *X*-axis, *Y*-axis, and *Z*-axis. In this case, the entire long maintenance cycle of bearings was about two years, indicating the degradation rate of bearings was slow. Thus, the intermittent sampling method was adopted. The sampling interval was set as 1 h, and sampling duration and sensor acquisition frequency were set as 1 s and 5000 Hz, respectively. The details of the experimental datasets are presented in Table 3.

Following the practical experience in numerous bearing studies, six bearing datasets were separated into a training set and a testing set to verify the effectiveness of the proposed approach. The training set accounted for about 70% and the testing set accounted for about 30%, which is also shown in Table 3.

In practical industrial environments, the degradation process of bearings has complex behaviors. As shown in Figure 7, the Bearing_1 dataset was evenly divided into 10 equal segments, and the boxplot method was employed to estimate amplitudes of vibration within each segment. It revealed a notable variation in the amplitude distribution over time, as well as significant differences among multiple axes, which also indicate the crucial potential of sequential characteristics and the fusion of vibration signals from various directions in predicting the RUL.

### 4.2. Experiment Setup and Evaluation Metrics

Due to different degradation mechanisms, the maximum operation cycles among bearing datasets usually have significant differences. Meanwhile, ensuring consistent scales between inputs and outputs of a model is advantageous for enhancing its prediction performance. Therefore, the RUL values of samples in this case were normalized in pretreatment.

Furthermore, to calculate prediction errors of approaches, an improved score function from the IEEE PHM 2012 challenge [23], mean absolute error (*MAE*) and root mean squared error (*RMSE*) were selected. The formulas for these metrics are defined as follows:(15)$$MAE=\frac{1}{N}\sum_{i=1}^{N}\left|{\tilde{y}}_{i}-y_{i}\right|$$
(16)$$RMSE=\sqrt{\frac{1}{N}\sum_{i=1}^{N}{\left({\tilde{y}}_{i}-y_{i}\right)}^{2}}$$
(17)$${Er}_{i}=\left(y_{i}-\tilde{y_{i}}\right)*100$$
(18)$$A_{i}=\left\{\begin{array}{c} e^{-{Er}_{i}*ln\left(0.5\right)/5},\;\;{Er}_{i}<0\\ e^{{Er}_{i}*ln\left(0.5\right)/20},\;\;{Er}_{i}\geq 0 \end{array}\right.$$
(19)$$Score=\frac{1}{N}\sum_{i=1}^{N}A_{i}$$
where N is the number of samples, and ${\tilde{y}}_{i}$ and $y_{i}$ are the *i*th predictive RUL value and actual RUL value, respectively.

### 4.3. Feature Construction

To obtain more comprehensive bearing degradation information, it is essential to construct multi-domain features (TDFs, FDFs, and TFDFs). In this case, TDFs consist of 16 statistical parameters and FDFs involve 12 statistical parameters. However, some statistical parameters are not sensitive to the degradation state of the bearing, which need to be eliminated to prevent negative effects. Therefore, monotonicity (*Mon*), correlation (*Corr*), as well as robustness (*Rob*) were considered for screening features [68], and a linear combination of these criteria was utilized as a comprehensive evaluation metric (*Cem*) to fully evaluate the applicability of degradation features. The formulas are outlined as follows:(20)$$\delta \left(k\right)=\left\{\begin{array}{c} 0,\;\;k<0\\ 1,\;\;k\geq 0 \end{array}\right.$$
(21)$$Mon\left(x\right)=\frac{1}{N-1}*\left|\sum_{i=1}^{N-1}\delta \left(x_{i+1}-x_{i}\right)-\sum_{i=1}^{N-1}\delta \left(x_{i}-x_{i+1}\right)\right|$$
(22)$$Corr\left(x\right)=\frac{\left|\sum_{i=1}^{N}\left(y_{i}-\bar{y}\right)\left(x_{i}-\bar{x}\right)\right|}{\sqrt{\sum_{i=1}^{N}{\left(y_{i}-{\bar{y}}_{i}\right)}^{2}*\sum_{i=1}^{N}{\left(x_{i}-{\bar{x}}_{i}\right)}^{2}}}$$
(23)$$Rob\left(x\right)=\frac{1}{N}\sum_{i=1}^{N}exp\left(-\left|\frac{x_{i}-{\hat{x}}_{i}}{x_{i}}\right|\right)$$
(24)$$Cem\left(x\right)=\frac{1}{3}*\left[Mon\left(x\right)+Corr\left(x\right)+Rob\left(x\right)\right]$$
where N is the number of samples; $y_{i}$ and $\bar{y}$ are the ith actual value and the mean value of RUL, respectively; $x_{i}$ and $\bar{x}$ are the *i*th value and the mean value of feature, respectively; and ${\hat{x}}_{i}$ denotes the *i*th smoothed value of feature.

Based on the training set, the results of comprehensive evaluation are shown in Figure 8, and the impact of parameters number on the proposed RUL prediction model are shown in Table 4. Finally, the threshold value of Cem is highlighted as the red line in Figure 8, the top eight statistical parameters of TDFs and FDFs were selected, which included:(1)Time domain parameters: RMS, mean, minimum, variance, clearance factor;(2)Frequency domain parameters: spectral mean, spectral root mean square, gravity frequency.

Additionally, the proposed CNN-VAE model has the capability to dynamically compress effective degradation information from the time–frequency spectrum into TFDFs. The compression effect of bearing datasets in the *X*-axis using the CNN-VAE model are presented in Figure 9.

As illustrated in Figure 9, the high-dimensional time–frequency spectrum of bearing datasets was efficiently compressed into nine TFDFs. The compression effectiveness of the CNN-VAE model is demonstrated in following aspects:(1)TFDFs curves of the Bearing_1 dataset in *X*-axis were closely related to the trend of the amplitude distribution over time in Figure 7. This means that the obtained TFDFs contained the important information of degradation process.(2)In addition, all TFDFs in the training set exhibited good monotonicity and robustness, which further proves the high performance of the CNN-VAE model in capturing important degradation information.(3)Moreover, it can be observed that the CNN-VAE model also had a successful performance in the testing set, and the extracted TFDFs had excellent continuity, indicating that the compression achieved by the CNN-VAE model demonstrates superior generalization.

### 4.4. RUL Prediction and Discussion

To validate the superiority of the proposed approach, this case sets up comparative experiments between the CNN-VAE-MBiLSTM model and four prediction models. These prediction models include linear support vector regression (LSVR), kernelized support vector regression (KSVR), DCNN, and BiLSTM. The main parameters and structures of these models are outlined below.

LSVR and KSVR: LSVR and KSVR are directly implemented using scikit-learn [69]. The penalty coefficients used for LSVR were set as 5, 50, 500, and 5000, respectively. The kernel trick for KSVR uses a radial basis function (RBF). Additionally, LSVR was utilized as the baseline model for comparison with other models.

DCNN: DCNN combines a CNN and a fully connected layer. The CNN part adopts the same structure as the CNN in the encoding part of the CNN-VAE model, as shown in Table 5. In addition, the fully connected layer is responsible for converting high-dimensional outputs from the CNN into prediction values of RUL.

BiLSTM: BiLSTM utilizes the same feature engineering as the proposed approach. But, in the prediction process, extracted features obtained from multi-axis were directly combined and then input into a single BiLSTM. The single BiLSTM follows the same structure as the BiLSTM part in the CNN-VAE-MBiLSTM model, in which the number of hidden layers and hidden neurons were set as 2 and 10, respectively.

Moreover, to ensure a fair and unbiased comparison, all models were evaluated using the same training set and testing set. All neural networks were trained with the Adam optimizer. The training epochs and the learning rate were set as 500 and 0.0005, respectively. To prevent contingency, the experiments were repeated three times, and those results were then averaged to obtain the final result. The experimental results are presented in Table 6 and Figure 10.

The detailed comparisons among these models are described as follows:(1)Based on TDFs and FDFs, both of the LSVR model and the KSVR model exhibited significant fluctuations in prediction results in the testing set. But, it is evident that the KSVR model outperformed the LSVR model in the training set, which indicates that better non-linear fitting ability is more conducive to establish an effective mapping relationship between features and bearing RUL.(2)Additionally, compared to the KSVR model, the MAE value and the RMSE value of DCNN model in the testing set decreased to 0.055 and 0.0883, respectively. This reduction of prediction errors is due to the utilization of additional TFRs, which also further verifies the significant impact of TFDFs in feature engineering.(3)Furthermore, the performance of the BiLSTM model was much better than the DCNN model, as shown in Figure 10. The MAE value and the RMSE value of the BiLSTM model in the testing set were 0.0414 and 0.0784, respectively. Meanwhile, TFDFs were adopted in the BiLSTM model to avoid the influence of TFRs. Thus, the difference between the BiLSTM model and the DCNN model reflects important effects of sequential characteristics.(4)Ultimately, the proposed CNN-VAE-MBiLSTM model integrates the extraction of TFRs and the obtainment of sequential characteristics. It is obvious that the proposed method achieves the best accuracy and robustness in RUL prediction. The values of MAE, RMSE, and Score in the testing set were 0.0281, 0.0401, and 0.7894, respectively, which means that the proposed approach can satisfy requirements of bearing maintenance in machines.

### 4.5. Robust Analysis

Generally, the noise in real industrial environments will result in a decrease in the predictive performance in various AI algorithms. Hence, the anti-noise capacity of algorithms plays a crucial role in determining its practicality. To evaluate anti-noise capacities of the proposed approach, different white Gaussian noises were added into the above comparative experiments. The intensity of white Gaussian noises was decided by the signal-to-noise ratio (*SNR*), as defined by:(25)$$SNR=10{log}_{10}\left(\sum_{i=1}^{n}A_{i,\;s}/\sum_{i=1}^{n}A_{i,\;n}\right)$$
where n is the signal length; $A_{i,s}$ and $A_{i,n}$ denote the *i*th amplitude value in raw data and white Gaussian noise, respectively; and the unit of *SNR* is decibel (dB).

In this section, the anti-noise capacity of the above algorithms was analyzed under different values of SNR that ranged from 10 dB to 2 dB. The lower SNR represents the higher intensity of the white Gaussian noise utilized. The experimental results are shown in Table 7, Table 8, Table 9 and Figure 11.

According to Figure 11, with the increase in the intensity of white noise, the accuracy of different models tends to decline. Specifically, when the SNR changed from 5 dB to 2 dB, there was a significant decreased in the accuracy of the LSVR model and KSVR model. Score values of both models in the testing set reduced from 0.4472 and 0.4127 to 0.2250 and 0.3202, respectively. In contrast, the accuracy of the DCNN model, BiLSTM model, and the CNN-VAE-MBiLSTM model declined more slowly; the scores of these models were 0.4636, 0.5437 and 0.6447 when the SNR was 2 dB. This suggests that the use of TFRs contributes to improving the robustness of models. Furthermore, when SNR was 2 dB, the proposed approach outperformed the other models, and achieved a further 18.6% improvement in Score compared to the BiLSTM model.

As discussed above, the proposed approach exhibited the best anti-noise capacity and the highest accuracy in noisy environment.

## 5. Conclusions and Future Research

In this paper, a novel approach for bearing RUL prediction called CNN-VAE-MBiLSTM is proposed. This approach can be divided into two parts: the CNN-VAE model and the MBiLSTM model. The CNN-VAE model is capable of automatically compressing the high-dimensional time–frequency spectrum of raw data into low-dimensional TFRs, which avoids laborious works of feature construction and eliminates the influence of personal participation. The MBiLSTM model adopts a two-step strategy that extracts features from each acquisition direction of a signal and will independently capture sequential characteristics at the first step. Following that, differences among multi-axis features are further obtained at second step. Ultimately, the proposed approach achieves accurate and robust RUL predictions.

In comparative experiments, the proposed CNN-VAE-MBiLSTM model was compared with four RUL prediction models (LSVR, KSVR, DCNN, and BiLSTM) to judge its prediction performance using three evaluation metrics. The comparison results confirmed the superiority of the proposed approach for RUL prediction. The MAE, RMSE, and Score of the proposed approach in the testing set were 0.0281, 0.0404, and 0.7894, respectively. In addition, the anti-noise capacity of the proposed approach was further analyzed by artificially adding different white Gaussian noises to the raw signals. As mentioned above, the proposed approach exhibited the best anti-noise capacity and the highest accuracy in a noisy environment.

In future research, the generalization of the proposed approach on different types of machines is planned to be discussed, and the network architecture of the proposed approach will be further optimized, aiming to achieve a better prediction performance and lower computational complexity.

## Figures and Tables

**Figure 1 sensors-24-02992-f001:**
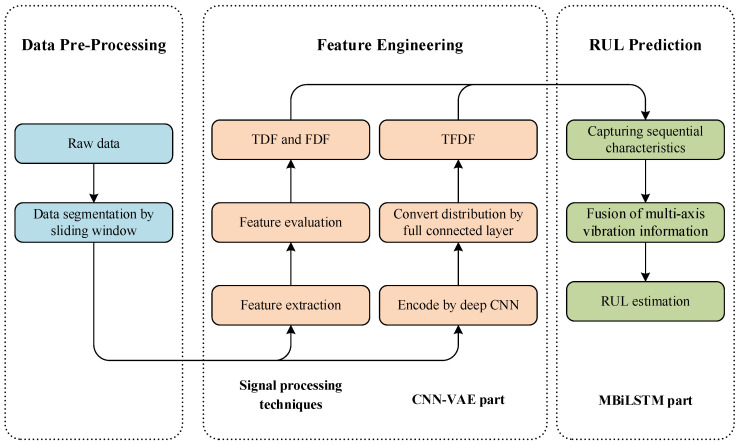
The diagram of the proposed approach.

**Figure 2 sensors-24-02992-f002:**
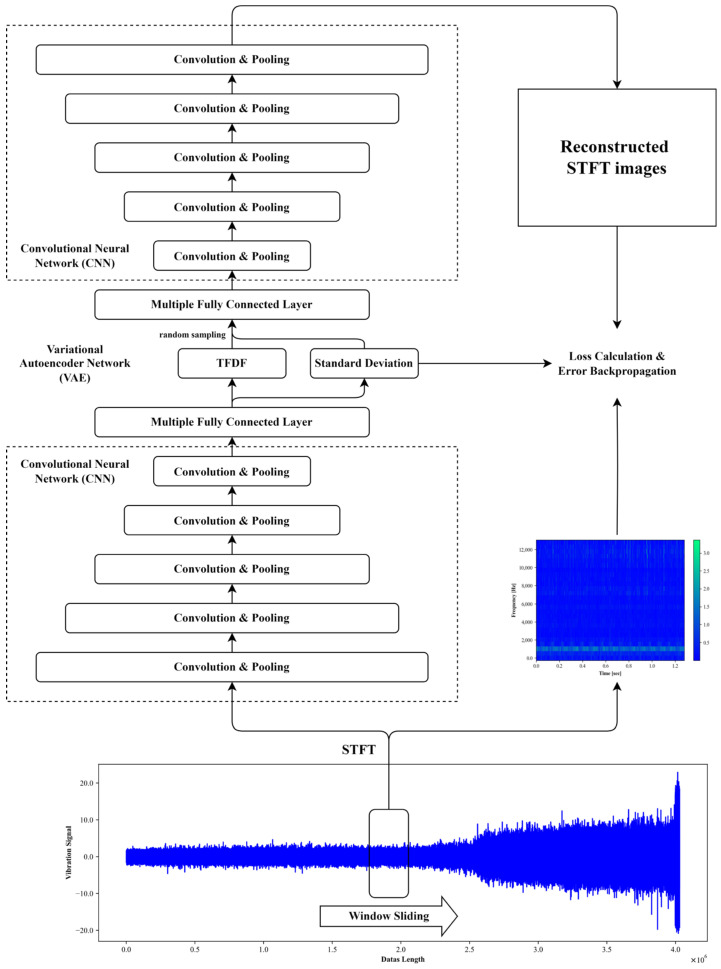
The schematic diagram of TFDF extraction.

**Figure 3 sensors-24-02992-f003:**
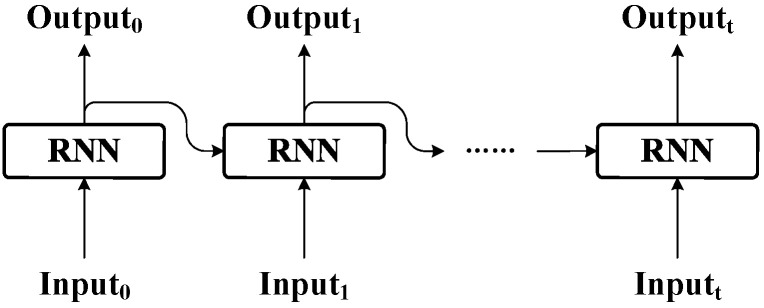
The architecture of RNN.

**Figure 4 sensors-24-02992-f004:**
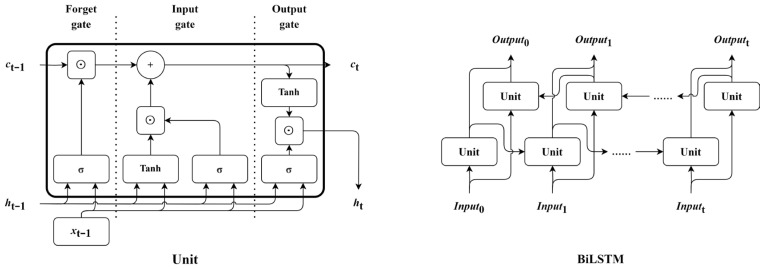
The structure of a neuron in the BiLSTM and the architecture of the BiLSTM.

**Figure 5 sensors-24-02992-f005:**
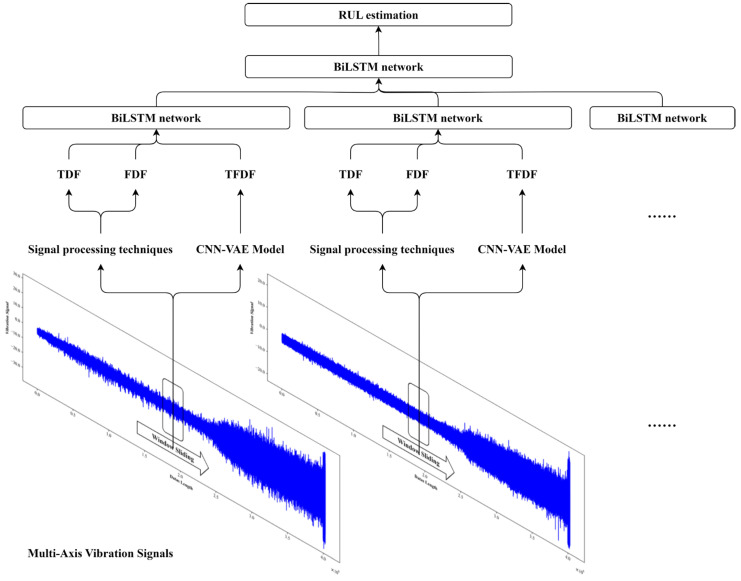
The schematic diagram of the CNN-VAE-MBiLSTM.

**Figure 6 sensors-24-02992-f006:**
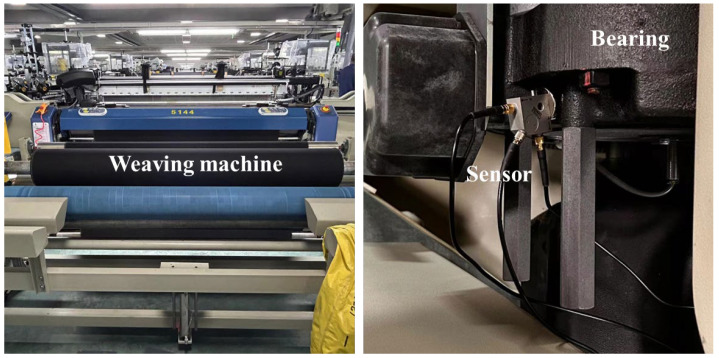
The experimental platform.

**Figure 7 sensors-24-02992-f007:**
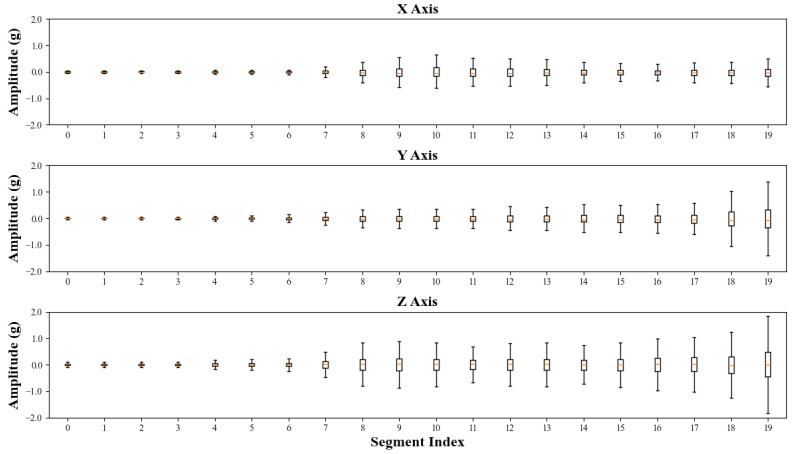
The amplitude boxplots of Bearing_1.

**Figure 8 sensors-24-02992-f008:**
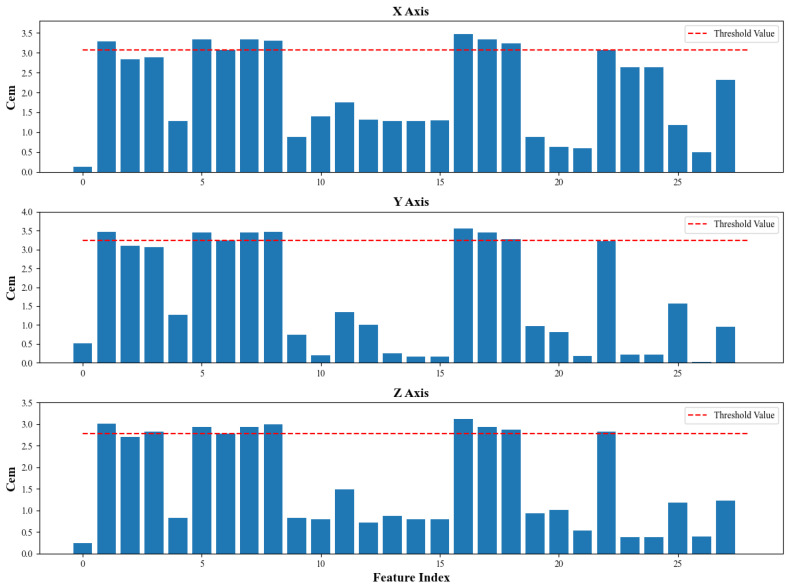
Comprehensive evaluation metric for TDF and FDF in multiple axes.

**Figure 9 sensors-24-02992-f009:**
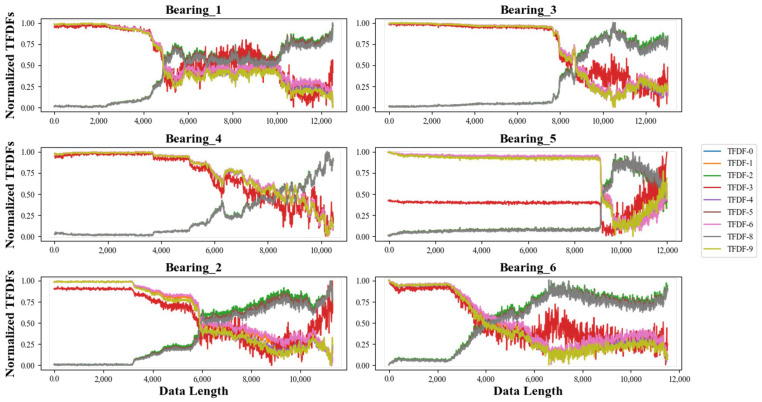
TFDFs of the training set and the testing set in the *X*-axis.

**Figure 10 sensors-24-02992-f010:**
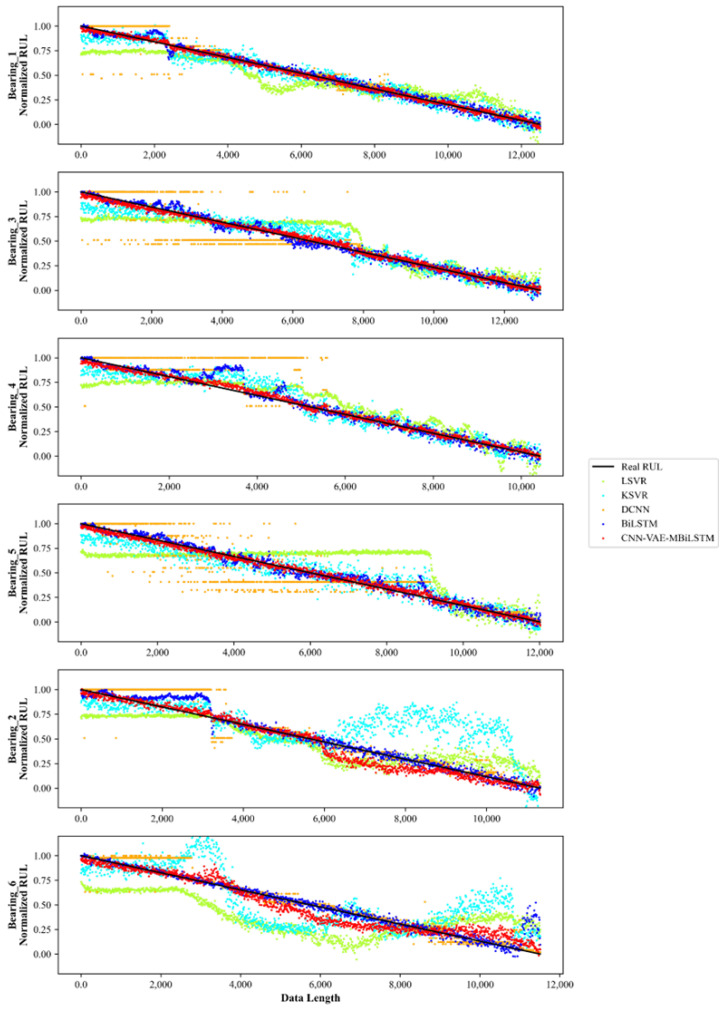
The RUL prediction performance of different models.

**Figure 11 sensors-24-02992-f011:**
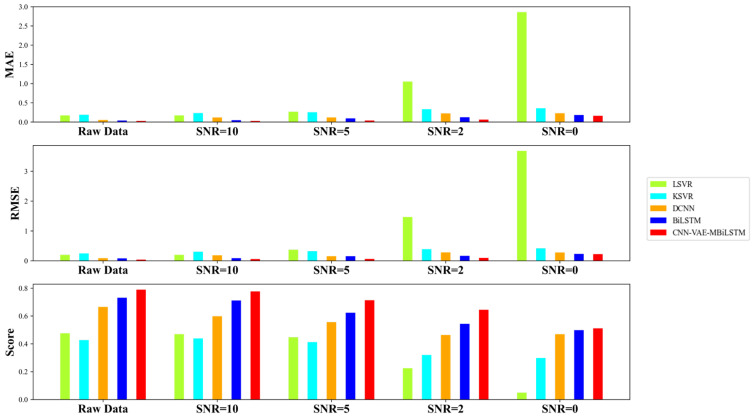
The performance of models under different background noise.

**Table 1 sensors-24-02992-t001:** List of TDFs.

**Name**	**Function**	Name	Function
Mean (μ)	$\frac{\sum_{i=1}^{N}x_{i}}{N}$	Absolute mean	$\frac{\sum_{i=1}^{N}\left|x_{i}\right|}{N}$
Minimum	$Min\left(x_{i}\right)$	Standard deviation (σ)	$\sqrt{\frac{\sum_{i=1}^{N}{\left(x_{i}-\mu \right)}^{2}}{N}}$
Maximum	$Max\left(x_{i}\right)$	$\mathrm{Variance}\;(\sigma^{2}$)	$\frac{\sum_{i=1}^{N}{\left(x_{i}-\mu \right)}^{2}}{N}$
RMS	$\sqrt{\frac{\sum_{i=1}^{N}{\left(x_{i}\right)}^{2}}{N}}$	Root square amplitude (RMA)	${\left(\frac{\sum_{i=1}^{N}\sqrt{\left|x_{i}\right|}}{N}\right)}^{2}$
Peak	$Max\left(\left|x_{i}\right|\right)$	Peak factor	$\frac{Max\left(\left|x_{i}\right|\right)}{RMS}$
Skewness	$\frac{1}{N}\sum_{i=1}^{N}{\left(\frac{x_{i}-\mu }{\sigma }\right)}^{3}$	Skewness factor	$\frac{Skewness}{{RMS}^{3}}$
Kurtosis	$\frac{1}{N}\sum_{i=1}^{N}{\left(\frac{x_{i}-\mu }{\sigma }\right)}^{4}$	Kurtosis factor	$\frac{Kurtosis}{{RMS}^{4}}$
Clearance factor	$\frac{Max\left(\left|x_{i}\right|\right)}{RMA}$	Impulse factor	$N*\frac{Max\left(\left|x_{i}\right|\right)}{\sum_{i=1}^{N}\left|x_{i}\right|}$

**Table 2 sensors-24-02992-t002:** List of FDFs.

**Name**	Function	Name	Function
$\mathrm{Spectral}\;\mathrm{mean}\;(\mu_{s}$)	$\frac{\sum_{i=1}^{M}w_{i}}{M}$	$\mathrm{Spectral}\;\mathrm{standard}\;\mathrm{deviation}\;(\sigma_{s}$)	$\sqrt{\frac{\sum_{i=1}^{M}{\left(w_{i}-\mu_{s}\right)}^{2}}{M}}$
Spectral skewness	$\frac{1}{M}\sum_{i=1}^{M}{\left(\frac{w_{i}-\mu_{s}}{\sigma_{s}}\right)}^{3}$	Spectral root mean square	$\sqrt{\frac{\sum_{i=1}^{M}{\left(w_{i}\right)}^{2}}{M}}$
Spectral kurtosis	$\frac{1}{M}\sum_{i=1}^{M}{\left(\frac{w_{i}-\mu_{s}}{\sigma_{s}}\right)}^{4}$	$\mathrm{Gravity}\;\mathrm{frequency}\;(f_{c}$)	$\frac{\sum_{i=1}^{M}w_{i}*f_{i}}{\sum_{i=1}^{M}w_{i}}$
Effective frequency	$\sqrt{\frac{\sum_{i=1}^{M}w_{i}^{2}*f_{i}^{4}}{\sum_{i=1}^{M}w_{i}}}$	$\mathrm{Standard}\;\mathrm{deviation}\;\mathrm{of}\;\mathrm{frequency}\;(\sigma_{f}$)	$\sqrt{\frac{\sum_{i=1}^{M}w_{i}*{\left(f_{i}-f_{c}\right)}^{2}}{\sum_{i=1}^{M}w_{i}}}$
Skewness of frequency	$\frac{1}{\sum_{i=1}^{M}w_{i}}\sum_{i=1}^{M}{w_{i}*\left(\frac{f_{i}-f_{c}}{\sigma_{f}}\right)}^{3}$	Mean square frequency	$\frac{\sum_{i=1}^{M}w_{i}*f_{i}^{2}}{\sum_{i=1}^{M}w_{i}}$
Variance of frequency	$\frac{\sum_{i=1}^{M}w_{i}*{\left(f_{i}-f_{c}\right)}^{2}}{\sum_{i=1}^{M}w_{i}}$	Variation coefficient of frequency	$\frac{\sigma_{f}}{f_{c}}$

**Table 3 sensors-24-02992-t003:** Details of experiment datasets.

Bearing Dataset	Number of Samples	Fault Element	Category
Bearing_1	12,514	Outer race	Training set
Bearing_2	11,323	Inter race	Testing set
Bearing_3	13,017	Inter race	Training set
Bearing_4	10,431	Outer race	Training set
Bearing_5	12,018	Cage	Training set
Bearing_6	11,510	Outer race	Testing set

**Table 4 sensors-24-02992-t004:** The impact of parameter number on the proposed RUL prediction model.

Parameter Number	Dataset	MAE	RMSE	Score
4	Training set	0.0302	0.0706	0.8343
	Testing set	0.0491	0.0916	0.7745
8	Training set	0.0104	0.0120	0.8693
	Testing set	0.0281	0.0401	0.7894
12	Training set	0.0298	0.0600	0.8449
	Testing set	0.0435	0.0795	0.7666
16	Training set	0.0329	0.0647	0.8277
	Testing set	0.0449	0.0813	0.7581
20	Training set	0.0330	0.0648	0.8273
	Testing set	0.0473	0.0843	0.7435
24	Training set	0.0357	0.0688	0.8119
	Testing set	0.0495	0.0870	0.7299
28	Training set	0.0466	0.0835	0.7477
	Testing set	0.0550	0.0932	0.6943

**Table 5 sensors-24-02992-t005:** The structure of the CNN part in DCNN.

Layer Name	Kernel	Strides	Channels	Feature Map Size
Convolutional Layer	3 × 3	(1,1)	16	128 × 512
MaxPool Layer	1 × 2	(1,2)		128 × 216
Convolutional Layer	3 × 3	(1,1)	32	128 × 216
MaxPool Layer	1 × 2	(1,2)		128 × 128
Convolutional Layer	3 × 3	(1,1)	64	128 × 128
MaxPool Layer	2 × 2	(2,2)		64 × 64
Convolutional Layer	3 × 3	(1,1)	128	64 × 64
MaxPool Layer	4 × 4	(4,4)		16 × 16
Convolutional Layer	3 × 3	(1,1)	256	16 × 16
MaxPool Layer	4 × 4	(4,4)		4 × 4

**Table 6 sensors-24-02992-t006:** The comparison of prediction ability among models.

Model	Feature Application	Dataset	MAE	RMSE	Score
LSVR	TDFs and FDFs	Training set	0.1198	0.1516	0.5064
		Testing set	0.1717	0.1998	0.4755
KSVR	TDFs and FDFs	Training set	0.0613	0.0770	0.6731
		Testing set	0.1880	0.2489	0.4270
DCNN	TDFs, FDFs, and time–frequency spectrum	Training set	0.0500	0.0878	0.7171
		Testing set	0.0550	0.0883	0.6655
BiLSTM	TDFs, FDFs, and TFDFs	Training set	0.0189	0.0361	0.8310
		Testing set	0.0414	0.0784	0.7312
CNN-VAE-MBiLSTM	TDFs, FDFs, and TFDFs	Training set	0.0104	0.0120	0.8693
		Testing set	0.0281	0.0401	0.7894

**Table 7 sensors-24-02992-t007:** The MAE of models under different white Gaussian noises.

Model	Dataset	Raw	SNR (dB)			
			10	5	2	0
LSVR	Training set	0.1198	0.1202	0.1664	0.6273	1.8530
	Testing set	0.1717	0.1734	0.2677	1.0532	2.8579
KSVR	Training set	0.0613	0.0685	0.1510	0.2455	0.3211
	Testing set	0.1880	0.2331	0.2550	0.3336	0.3599
DCNN	Training set	0.0500	0.0863	0.1354	0.1603	0.1587
	Testing set	0.0550	0.1198	0.1219	0.2247	0.2304
BiLSTM	Training set	0.0189	0.0322	0.0894	0.1136	0.1529
	Testing set	0.0414	0.0494	0.0963	0.1235	0.1829
CNN-VAE-MBiLSTM	Training set	0.0104	0.0183	0.0380	0.0818	0.0800
	Testing set	0.0281	0.0300	0.0400	0.0632	0.1635

**Table 8 sensors-24-02992-t008:** The RMSE of models under different white Gaussian noises.

Model	Dataset	Raw	SNR (dB)			
			10	5	2	0
LSVR	Training set	0.1516	0.1535	0.2166	1.1201	3.2188
	Testing set	0.1998	0.2021	0.3691	1.4647	3.6824
KSVR	Training set	0.0770	0.0862	0.1925	0.2896	0.3670
	Testing set	0.2489	0.3017	0.3204	0.3882	0.4162
DCNN	Training set	0.0878	0.1400	0.1991	0.2142	0.2019
	Testing set	0.0883	0.1821	0.1557	0.2817	0.2788
BiLSTM	Training set	0.0361	0.0525	0.1444	0.1694	0.2083
	Testing set	0.0784	0.0892	0.1532	0.1668	0.2317
CNN-VAE-MBiLSTM	Training set	0.0120	0.0342	0.0694	0.1367	0.1232
	Testing set	0.0401	0.0584	0.0630	0.0985	0.2209

**Table 9 sensors-24-02992-t009:** The Score of models under different white Gaussian noises.

Model	Dataset	Raw	SNR (dB)			
			10	5	2	0
LSVR	Training set	0.5064	0.5035	0.4807	0.3530	0.1374
	Testing set	0.4755	0.4699	0.4472	0.2250	0.0488
KSVR	Training set	0.6731	0.6554	0.5207	0.3968	0.3283
	Testing set	0.4270	0.4383	0.4127	0.3202	0.2981
DCNN	Training set	0.7171	0.6384	0.5751	0.5145	0.5021
	Testing set	0.6655	0.5978	0.5567	0.4636	0.4695
BiLSTM	Training set	0.8310	0.8047	0.6937	0.6423	0.5628
	Testing set	0.7312	0.7110	0.6238	0.5437	0.4983
CNN-VAE-MBiLSTM	Training set	0.8693	0.8449	0.7798	0.7126	0.6151
	Testing set	0.7894	0.7765	0.7132	0.6447	0.5109

## Data Availability

The data are not publicly available due to experimental privacy.

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
