# Peer review of "Rolling Bearing Remaining Useful Life Prediction Based on CNN-VAE-MBiLSTM"

_sensors, 2024, doi:10.3390/s24102992_

Round 1

Reviewer 1 Report

Comments and Suggestions for Authors

In this manuscript, a method is proposed to predict the Remaining Useful Life of rolling bearings by integrating CNN, VAE, MBiLSTM. This manuscript is well-organized with associated results clearly presented. Specific comments are as follows:

(1) Abbreviations should only be expanded upon their first occurrence in the text. For example, if "RUL" is defined in the Introduction, there is no need to provide its full name again at line 35 of the document. Please check and correct similar issues throughout the manuscript.

(2) In Section 1, lines 49-52, Reference 16 is mentioned as pertaining to data-driven methods. However, the described content in this section actually relates to mechanism model methods. Please review this part and ensure that the referenced literature is appropriately matched to the described content.

(3) In Table 2, the fifth row is slightly narrow, causing the formula within it to be displayed incompletely. Please correct this issue. Additionally, the meaning of "fc" in the formula under the parameter name "Variation coefficient of frequency" within the table is not clearly explained. Clarification is required.

(4) In Section 3.1, what are the sizes of the sliding windows and the amount of slide? How were these determined? Please clarify these details in the text.

(5) In Section 4.3, what criteria were finally established by Cem? It is not clear what the final selection of eight parameters are and how they were chosen. This needs to be explicitly stated.

(6) More recent works should be included in this manuscript, like “A novel machine learning method for multiaxial fatigue life prediction: Improved adaptive neuro-fuzzy inference system. International Journal of Fatigue, 2024, 178: 108007.”, “Fatigue reliability analysis of composite material considering the growth of effective stress and critical stiffness. Aerospace, 2023, 10(9): 785.”

(7) Consider reformatting Figure 10 to ensure that the text and content within the figure are clearly legible. In Figure 11, ensure that the legend does not obscure any information of the bar charts.

       (8) Section 1 also mentions some Related Work. Consider integrating this into Section 2 for coherence and consolidation.

Reviewer 2 Report

Comments and Suggestions for Authors

In this work is proposed a method based on CNN-VAE-MBiLSTM applied for the estimating the remaining useful life of bearing elements, the proposal is interesting and the results are promising.

Some issues have to be addressed in order to highlight the contribution.

1. As an original research work, it is suggested to reduce the percentage of similitude since the actual percentage is around 32%, the authors have to reduce those sections that are similar to other works.

2. The state-of-the-art can be completed by including some works that have been proposed in the field of fault detection in bearings elements but bearings made with different materials, for example, ceramic and hybrid bearings. The following works may be taken into account: https://doi.org/10.1016/j.engfailanal.2023.107213 ; https://doi.org/10.3390/s21175832 ; https://doi.org/10.1007/s12555-021-0167-0

3. The proposal includes the calculation of time and frequency domain features, are there significant differences whether only TDF or TFF are used as the fault related features? Can the authors include a comparison between TDF, TFF and both (together)?

4. What are the disadvantages or limitations of the proposed method? Please include a brief discussion in the conclusion section.
